# Effects of Varied Stimuli on Escape Behavior Diversification of Himalayan Marmots for Different Human Disturbances

**DOI:** 10.3390/ani15070935

**Published:** 2025-03-25

**Authors:** Tao Lei, Hua Peng, Han Zhang, Ying Ban, Muhammad Zaman, Zuofu Xiang, Cheng Guo

**Affiliations:** 1College of Life and Environmental Sciences, Central South University of Forestry & Technology, Changsha 410004, Chinazhanghan2608@163.com (H.Z.); 2Institute of Evolutionary Ecology and Conservation Biology, Central South University of Forestry & Technology, Changsha 410004, Chinaxiangzf@csuft.edu.cn (Z.X.); 3Administration Bureau of Sichuan Ruoergai Wetland National Nature Reserve, Ruoergai, Aba 624500, China; 4College of Forestry, Central South University of Forestry & Technology, Changsha 410004, China

**Keywords:** approaching style, escape behavior, Himalayan marmot, human disturbance

## Abstract

Many studies have found that wild animals exhibit a longer FID when threatened by a single dog or a leashed dog accompanied by a human, i.e., dogs are treated as a greater threat than humans. This study investigated the effects of different stimuli (a running or walking man with or without a leashed dog and a dog alone) on the escape behavior of four Himalayan marmot populations experiencing different human disturbances. Surprisingly, these Himalayan marmots exhibited a unique escape strategy relative to other marmot species. Overall, they treated humans as a greater threat than dogs: the alert distance (AD) and flight-initiation distance (FID) derived from dogs were shorter than the AD and FID derived from humans and were unaffected by the intensity of disturbance. When the stimuli involved humans, the AD and FID both become shorter with an increase in the disturbance intensity. Moreover, the effect of threat speed on the escape behavior was evident only in the population disturbed at the highest sites (a faster speed and a longer FID). These new study findings may result from the small size and white color of the dog used in this study. To further explore the effect of dogs on the escape behavior of Himalayan marmots, different-sized dogs with varied colors should be utilized in future studies.

## 1. Introduction

Disturbances from human activities are increasingly having a negative impact on wildlife [1,2]. These disturbances can affect multiple aspects of wildlife like behavior, inter-specific relationships, and physiological or morphological characteristics [3,4,5,6]. To some extent, populations experiencing disturbances from human activities can be adaptive toward these disturbances [7,8], and behavioral adaption (i.e., adjusting time budget and rhythm, shifting the way of habitat utilization, etc.) is an efficient adaptive strategy for different kinds of human disturbances [9,10,11].

Among the strategies used to deal with potential dangers (both from humans and natural predators), escaping far away from the approaching threat is the most common and most studied strategy [12]. The escape response can be directly illustrated by three parameters: the alert distance (the distance between an animal and an approaching potential predator at which point the animal begins to exhibit alert behaviors to the approaching potential threat, AD); flight-initiation distance (the distance between an animal and a threatening stimulus at which it begins to escape, FID); and distance fled (the distance between the flight start point and the final safe location at which the threatened animal arrives after escaping, DF) [13,14]. Meanwhile, the buffer distance (BD), the difference between the AD and FID, which illustrates the sensibility of animals facing the approaching threat, is another important parameter to measure escape behavior [15,16].

Previous studies indicated that, when populations of the same animal species were disturbed by different disturbances or threats, an intra-specific adaptive strategy of diversification exists (exhibited in the form of different ADs, FIDs, BDs, and DFs), depending on the intensities or kinds of disturbances or threats. For example, the approaching style of the approaching threat (walking or running human or vehicle) toward a focal animal and the environmental background (an open or a closed field) may trigger different escape reactions [17]. Forest elephants (*Loxodonta cyclotis*) endemic to Gabon are more sensitive to and, consequently, actively avoid humans that are passing by (longer ADs and FIDs), but they are relatively more relaxed when disturbed by the high sound of dynamite blasts [5]. The triggers of escape of both black-tailed deer (*Odocoileus hemionus*) and white-tailed deer (*O. virginianus*) are faster when threatened by humans in an open field than in a closed field (longer FIDs), and they are more sensitive to passing-by walking humans (longer ADs) than to passing-by vehicles [18]. When threatened by a walking dog (*Canis lupus familiaris*), free-living, gregarious sparrows (Passerellidae) first evaluate the threat level and then decide the step to be taken, but they fly off immediately when approached by a running dog [19]. Light-vented bulbuls (*Pycnonotus sinensis*) at higher perched locations have longer BDs, and this species tend to delay their escape after detecting an approaching predator when perched high [15].

Furthermore, different disturbed populations may react differently (exhibit varied ADs, FIDs, and DFs) when experiencing the same disturbance. Blue-tailed skinks (*Emoia impar*) tolerated a closer approach and fled a shorter distance in locations with a relatively low human disturbance than in locations with a medium or a high human disturbance, indicating that skinks are more sensitive in disturbed areas [14]. Despite low threats from native predators, house finches (*Carpodacus mexicanus*) in highly urbanized habitats had a greater tendency to escape to larger distances than their congeners in less urbanized habitats when threatened by the same threat [20]. Conversely, Himalayan marmots (*Marmota himalayana*) in highly disturbed habitats have a shorter FID relative to their congeners in less disturbed habitats [7]. Foraging Olympic marmots (*M. olympus*) in highly disturbed sites tolerated more human contact, but they also looked around more often (which suggested an increased wariness) [21].

Dogs, like domesticated wolves, present in or around cities or villages usually prey on and consequently pose more threat than humans to sympatric wild animals [22]. For example, urban burrowing owls (*Athene cunicularia*), Alpine marmots (*M. marmota*), and long-tailed marmots (*M. caudata*) recognize humans with dogs as greater threats than humans alone: an approaching dog triggers their escape reaction and obviously enhances the FID of the disturbed target individuals [23,24,25], and predation on sympatric species by stray dogs generally results in the population loss of these sympatric animals [26].

As residents of the Qinghai–Tibetan Plateau, Himalayan marmots must face the challenges of ambient seasonal and harsh environmental conditions [27,28]. To gain enough weight to survive a long hibernation period [29], Himalayan marmots disturbed by the daily activities of yaks, sheep, and humans adjusted their time rhythm (foraged more at noon) [11] and reduced their FID relative to undisturbed individuals to maintain their feeding efficiency [7]. However, how the escape reaction of Himalayan marmots experiencing different human disturbances diversified when disturbed by different stimuli (i.e., a walking or running man with or without a leashed dog or a single dog as a stimulus) needs further exploration [24,25,30]. To answer this question, Himalayan marmots belonging to four populations experiencing persistent but different intensities of human disturbances in the northeastern Qinghai–Tibetan Plateau were threatened by five stimuli: a dog alone, a human alone, a dog leashed by a human, a running human, and a walking human [7,30,31]. The AD, FID, BD, and DF of each marmot individual were recorded and comparatively analyzed to explore the following: (1) inter-population variation when marmots were disturbed by the same threat; (2) intra-population variation when marmots were disturbed by different threats; and (3) further analyzing the adaptive strategy of the species experiencing varied intensities of human disturbances pertaining to different kinds of potential threats. Moreover, we predict the following in this study: (1) when disturbed by varied stimuli, intra-population variations emerge and longer ADs and FIDs are observed in dog-derived than in human-derived escape reactions; furthermore, running can result in a longer AD and FID than walking; (2) when disturbed by the same stimulus, the general tendency is that the AD and FID become shorter with an increase in the disturbance intensity.

## 2. Materials and Methods

### 2.1. Study Site

This study was conducted around a village (Duoma) in the Zoige wetland (103.01°E, 33.5°N), northeastern Qinghai–Tibetan Plateau. The village is about 8.5 km southeast of Zoige County [32]. Compared with a previous study conducted in 2020 [7], with the renovation of the road to the county that bypasses the village in 2021, two makeshift roads that pass through the former highly disturbed habitat have been abandoned (Figure 1). Meanwhile, with the construction of a railway across the study area since late 2022 and the refurbishment of the makeshift road in the wetland in 2023 (Figure 1), the intensities and sources of human disturbances experienced by local Himalayan marmots have changed significantly compared with those in 2020 [7].

### 2.2. Disturbance Intensity

Based on previous studies, the intensity of human disturbances that different populations experienced was re-quantified based on the number of passing vehicles [7,31]. Furthermore, since there may be a divergence in the marmots’ perception of danger to different types of vehicles (i.e., trucks, cars, motorcycles, etc.) [17], we quantified the disturbance intensity of different vehicle types: motorcycles, electric bicycles, and horses were regarded as 0.5 units, cars were regarded as 1 unit, small- and medium-sized trucks were regarded as 1.5 units, and large trucks were regarded as 4 units.

Daily records from 8:00 AM to 6:00 PM were gathered every 36 days across four regions, and there was an obvious divergence observed in the disturbance intensity (Figure 1): (1) the high disturbed district (HDD) had a certain area overlap with the previous high disturbed district; (2) the medium disturbed district (MDD) was the previous low disturbed district; (3) the low disturbed district (LDD) comprised two regions along the road to summer pasture; and (4) the natural district (ND) overlapped with the previous natural area [7,31].

### 2.3. Data Collection

We carried out 600 sets of simulations using different stimuli and collected escape responses from four marmot habitats from mid-July to early September 2023. A total of 150 sets of approaching (a running or walking human with or without a dog and a single dog) followed by the approached marmot’s alerting and escaping were conducted across each habitats. In each habitat, 30 sets of data had solely a dog as the stimulus, 30 sets of data had solely a man walking or running as the stimulus (60 sets of data in total), and 30 sets of data had a running or walking man with a leashed dog as the stimulus (60 sets of data in total). The corresponding AD, FID, and DF of the focal marmots were then recorded, and the BD was calculated as the AD minus the FID. Individuals of 35 reproductive pairs (2–6 adult individuals in each pair) from the HDD were threatened, and the four derived distances were measured. A reproductive pair was subjected to an average of 4.3 experiments because it is hard to identify different individuals; in the most extreme conditions, the same individual from the same pair was on average selected and tested 4.3 times, and the interval of each test was at least ten days. Each experiment performed on the same pair can be considered as independent. Furthermore, the number of reproductive pairs in the MDD, LDD, and ND was significantly more than 35; consequently, all the experiment cases were considered independent.

### 2.4. Experimental Procedure

As in the previous study [7], this study’s sample was arranged as follows: Firstly, an adult marmot from a random reproductive pair (determined by its burrow location) was selected as the target in a certain habitat (Figure 1), and its current state (foraging, resting, or alerting) was recorded. Then, one researcher (Lei Tao, 180 cm in height (Appendix A), wearing a green jacket and black trousers throughout the experimental period; hereafter, he is known as the experimenter) approached the target marmot alone in its line of sight. Meanwhile, the recorder (Peng Hua) observed and recorded the approaching style (walking or running and the existence of the leashed dog (an adopted white hybrid stray dog with wildlife experience, 40 cm in height, Appendix A)) of the experimenter, and the reaction of the target marmot was observed from a distance (usually more than 100 m) with a telescope (BOSMA, 20–60 × 80 and OLYMPUS, 10 × 42). Both the approach and the observation were conducted carefully to avoid the premature escape of the marmot.

During the approach, the experimenter also observed and recorded the state shift (i.e., foraging to alerting, alerting to escaping, etc.) of the focal marmots, and he dropped his gloves to mark his spots when the marmot started to alert and escape. Furthermore, a third glove was used to mark the position where the marmot started to escape from.

Then, the experimenter measured the number of steps between the three gloves: the distance between the third glove and the first glove is the AD, and the distance between the third glove and the second glove is the FID. The DF is the distance from the third glove to the burrow that the focal marmot entered (only straight escape were considered). The BD is the difference between the AD and the FID.

In the experiments involving a single dog, it was necessary to guide the dog to identify and approach the target marmot in advance (to avoid the overlap with simulations that involve a man with a leashed dog as the potential predator), and the experimenter only needed to stand at a distance to observe and record the actions of the dog and the reactions of the focal marmots.

### 2.5. Data Analysis

We used a *T*-test in GraphPad Prism 8.0.1 to re-identify the disturbance intensity of different marmot populations around Duoma village, and this resulted in four regions with different intensities (Figure 1). Then, we analyzed the diversification of varied stimuli and different human disturbance intensities that produced four escape-derived distances (the AD, FID, BD, and DF). One-way ANOVA was also used with GraphPad Prism 8.0.1 to analyze whether significant intra-population differences exist when marmots experience the same disturbance when threatened by different stimuli and whether significant inter-population differences exist when marmots experiencing varied intensities of human disturbances are threatened by the same stimulus. Furthermore, the unpaired *T*-test was used to compare whether significant differences existed between the five stimuli (walking man, running man, walking man with a leashed dog, running man with a leashed dog, and a dog alone).

## 3. Results

### 3.1. Intra-Population Variation When Disturbed by Varied Stimuli

No significant difference in the AD emerged when marmots were threatened by varied approaching styles (walking man, WM; running man, RM; walking man with a leashed dog, WM&D; running man with a leashed dog, RM&D; and a dog alone, DA) in the MDD (F = 0.1446; *p* = 0.9653) (Figure 2B).

In the HDD, no diversification emerged when threats involved a human and when threats involved a dog. Nevertheless, the single dog resulted in a longer AD than a man alone (Figure 2A). Both an individual and the human–dog combination produced similar variations in the AD for marmots from the LDD and ND; however, these ADs were longer than the ADs resulting from the dog (Figure 2C,D).

The FID was generally longer when an individual or the human–dog combination was used as a stimulus than that for a dog alone across the four populations, though a dog alone as a stimulus resulted in no significant differences from the WM and WM&D in the HDD (Figure 2E). In the HDD, the FIDs of a walking human (t = −9.165; *p* = 0.0027) or human–dog combination (t = −15.26; *p* = 0.0013) were shorter than the FIDs observed when they were running toward the focal marmots (Figure 2E). Nevertheless, the approaching style involving a human resulted in similar reactions from marmots across the MDD, LDD, and ND (Figure 2F–H).

A single dog resulted in a longer BD relative to a sole man or a human–dog combination across four habitats (in addition to the walking human–dog combination in the LDD and ND). In the MDD, LDD, and ND, human-derived stimuli showed no diversification (Figure 3J–L); nevertheless, a running man resulted in a shorter BD than a walking man in the HDD (Figure 3I).

The approaching style had no effect on the DF when a man alone or human–dog combination was the potential threat, but a dog-derived threat resulted in a shorter DF compared with a man alone (Figure 2M–P).

### 3.2. Inter-Population Variation When Disturbed by the Same Stimulus

When approached by a man or human–dog combination (walking or running), significant differences in the alert distance were observed in the four habitats (WM: F = 89.62; *p* < 0.0001; RM: F = 71.78; *p* < 0.0001; WM&D: F = 41.34; *p* < 0.0001; and RM&D: F = 34.58; *p* < 0.0001). The AD generally decreased with an increase in the disturbance intensity when disturbed by a man or a human–dog combination (Figure 3A–D). Nevertheless, a single dog as a stimulus resulted in no differences in the AD across individuals from different areas (F = 0.1813; *p* = 0.9096) (Figure 3E).

Across the four regions, the variations in FID were similar to the diversification of AD when threatened by a man or a human–dog combination (WM: F = 87.91; *p* < 0.0001; RM: F = 52.87; *p* < 0.0001; WM&D: F = 34.76; *p* < 0.0001; and RM&D: F = 23.86; *p* < 0.0001), and the FID generally decreased with an increase in the human disturbance intensity (Figure 3F–I). Similarly, a single dog as a stimulus resulted in no differences in the FID across the four districts (F = 0.2901; *p* = 0.8325) (Figure 3J).

A walking man, a single dog, and a running human–dog combination resulted in no diversification in the BD across the four regions (Figure 3K,N,O). Diversification emerged when a running man resulted in a longer BD in the ND than in the other three habitats (Figure 3L) and when the walking human–dog combination resulted in the shortest BD in the MDD (Figure 3M).

Significant differences in the DF were detected only when the walking human–dog combination approached the target marmots (F = 3.293; *p* = 0.0229) (Figure 3R). No difference was detected in the DF when the stimulus was a walking man (F = 1.975; *p* = 0.1174), a running man (F = 0.4501; *p* = 0.7175), a running human–dog combination (F = 1.660; *p* = 0.1799), and a single dog (F = 0.1448; *p* = 0.9328) across all four populations (Figure 3P,Q,S,T).

## 4. Discussions

Diversification was observed in the alert distance of marmots from the HDD, LDD, and ND when they were threatened by varied stimuli. Generally, human-derived stimuli show no significant effect on the AD across four habitats, and, especially in the LDD and ND, stimuli related to a human result in the same AD, and all ADs associated with a human were longer than the ADs derived from a single dog. Thus, it can be inferred that the marmots from those two populations were more alert toward humans than dogs, and this divergence may be caused by the small-sized dog used in this study and, consequently, by the lower visibility relative to the experimenter (40 cm vs. 180 cm; Appendix A). Moreover, the grass was about 20 cm to 40 cm in height during this study; therefore, the grass was an ideal cover for the approaching dog, and it was detected by the focal marmots only at close distances. In the LDD and ND, compared with small-sized dogs, humans are more likely to be noticed by marmots, which results in a longer AD [12,33], and the state (running or walking, with or without a dog) of the approaching human is not very important for the AD of marmots.

Marmots from the HDD alert more when foraging (data from a field observation conducted in 2024), enabling them to detect potential threats in a timely manner, and they exhibit similar ADs when threatened by a dog or a human. Interestingly, no variation emerged in the AD of marmots from the MDD when they were threatened by different stimuli, probably because they are always alert and can detect various potential threats in a timely manner.

Fatal disturbances (i.e., hunting) increase the AD (and the FID) of many ungulates [34,35]. The disturbances assessed in this study are nonfatal, and the marmots gradually habituated to these disturbances. The divergence in AD may result from the different levels of habituation. Marmots from the LDD have been disturbed since 2022, but with a low level of habituation, they are more tolerant than marmots from the ND, consequently, exhibiting a relatively shorter AD (76.0 m vs. 95.3 m). Similar to yellow-bellied marmots (*M. flaviventris*) and Olympic marmots, marmots from the HDD are the most habituated and also the most alert, enabling them to quickly detect and respond to potential threats [20,36,37]. Marmots from the LDD are less habituated [7], but they are also more alert, a habit that helps them detect small-sized predators in the grassland (i.e., the dog in this study). It was concluded that intra-population diversification pertaining to the AD, when the marmots are threatened by various stimuli, may be determined by the alert intensity and visibility of threats.

There is another possibility that marmots from the HDD had already noticed the impending threats before alerting and that they postponed the alert due to a high level of habituation and then began to alert at a certain distance. In contrast, marmots from the other three habitats started to alert immediately upon detecting the approaching threat, i.e., they showed a high sensitivity to threats.

Generally, wild animals treat dogs or the human–dog combination as a greater threat than a single human (a longer AD and a longer FID) [22,23,25,38]. Nubian ibex (*Capra nubiana*) exhibit a longer AD when threatened by a man with a leashed German shepherd (black and yellow in color and about 65 cm in height) than that for a man alone [39]. This study did not find similar results may be due to the different color and size of the dog used in this study; a white dog that is 40 cm in height is a new threat for the studied marmots and requires more time for identification.

After detecting the approaching predator and starting to alert, the movement of the predator triggers the escape reaction in marmots, similar to the AD, and FIDs derived from a single dog were shorter in the MDD, LDD, and ND at the intra-population level. For the three relatively low-habituated populations, big-sized humans are more likely to be detected no matter what their approaching style (running or walking, with or without a dog). This observation is different from those of all other studies involving dogs, which found that a single dog or a human with a leashed dog resulted in a longer FID than only a man [22,23,24,38,39,40]. This difference in observation may also result from the small-sized and white-colored dog used in our study, as no natural predator of the marmots has the same size and color characteristics, which may have hindered the marmots in identifying the approaching threat and conducting the next steps.

In the HDD, a walking man (with or without a dog) resulted in the same FID as a single dog, and this may be because marmots at high habituation levels are more alert but less sensitive to disturbances than their congeners from the other three populations. As these marmots have experienced long-term nonfatal disturbances from dogs, humans, and vehicles, they have similar risk perception thresholds for different disturbances. A running man as a stimulus shortened the FID of marmots in the HDD, indicating that these marmots are sensitive to the speed of approaching threats and escape earlier to avoid being preyed on by fast predators [29,41].

At the inter-population level, stimuli involving humans resulted in a trend of the FID becoming longer with the decrease in the disturbance intensity; this trend indicates that the more the disturbances experienced by the marmots, the more sensitive the marmots become to these disturbances [36,41,42]. The most disturbed marmots tolerate a closer approach by a human (walking or running, with or without a dog), and this is their adaptive strategy to human disturbances [7,39]. When approached by a single dog, the FID of marmots across all four populations showed no diversification, indicating that, though they experienced varied intensities of disturbances, the marmots from all the populations evaluated a 40 cm tall white dog as a threat on the same level and, consequently, showed the same reactions.

Contrary to other studies, we found that the dog-derived FID was shorter than FIDs derived from a man alone or a man accompanied by a dog and was independent of the disturbance intensity [22,23,24,39]. This divergence may result from the small-sized dog used in this study, and this dog was treated as less dangerous by the studied marmots; consequently, even the most-sensitive marmots from the MDD showed a shorter dog-derived FID than the human-derived FID.

Previous studies have found that humans with different clothes influence the escape reaction of birds [43,44]; consequently, in addition to the body size, the color of the dog in this study, i.e., white, may also contribute to the shorter dog-derived FIDs of the studied marmots. There are no white natural predators of marmots in the region; thus, marmots that are being approached can only identify the approaching dog at closer distances, resulting in a shorter FID and a longer buffer distance (BD, see below). Together with other studies [45], we assumed that marmots react differently to different-sized and different-colored predators (i.e., different reactions to a fox and a wolf).

Generally, dogs resulted in a longer BD than BDs derived from human-accompanied dog stimuli, and this pattern indicated that marmots spent more time evaluating the danger level when facing an approaching dog. This pattern is also different from those reported by other studies [23,24]. When facing a white dog with a height of 40 cm, marmots need more time to evaluate when they should escape. In this study, the marmots treated the dog as a lesser threat than a 180 cm human (and the human–dog combination), and they may react differently when facing a large-sized dark dog [39].

Human-accompanied dog stimuli resulted in no variation in the BD among marmots from the MDD, LDD, and ND; such a consistent reaction may result from the marmots of these populations being less alert but more sensitive and reacting similarly to human-accompanied dog stimuli from the same evaluation distance. Nevertheless, marmots from the HDD treated the running man (with or without a dog) as a greater threat than the walking man (with or without a dog). These marmots can adjust their escape strategy according to the speed of approaching threat.

Combined with the diversification of ADs, FIDs, and BDs, when a human is a part of a disturbance, marmots always detect the human first. Future studies will consider a bigger dog with a darker color in diverse seasons (shorter grass) to better explore the escape reactions of marmots to different-sized predators.

The distance fled depends on the foraging location and the location of the shelter burrow [7]; there is selective divergence pertaining to the shelter burrow when marmots evaluate the approaching threat and start to escape. Individuals from all four populations showed a trend that the DF was longer when a man alone was the threat. The diversification may result from two opposite reasons: (1) a threat from the human–dog combination is the same as that from a single dog and more dangerous than a man alone, and marmots preferentially enter into the nearest burrow; nevertheless, when a man alone is less dangerous, they choose to enter a farther burrow; (2) a threat from a human–dog combination is the same as that from a single dog and less dangerous than a man alone; marmots preferentially enter into the nearest burrow and reemerge back to the ground to guarantee their feeding efficiency [7]. Contrarily, when facing the more dangerous individual, marmots preferentially enter further but safer shelter burrows and stay for a longer time. The reasons for why the studied marmots treated a human–dog combination and a single dog as similar threats and how they selected shelter burrows need further investigation.

At the inter-population level, in addition to a walking man with a leashed dog resulting in the diversification of DF, the DF was undetermined for all other stimuli, indicating that, though disturbed differently, marmots from different populations had the same risk perception threshold for the same threat.

In similar studies involving dogs, excluding the study on Nubian ibex that illustrated the color and size of their dog in a figure [39], other studies that considered dogs as a threat generally did not provide the descriptions of the size and color of the dog used [22,23,24,38]. However, the color and size of a threat may greatly influence the reaction of individuals experiencing a threat [46,47], and we strongly recommend that future studies take the physical characteristics of the dog used into account.

## 5. Conclusions

The Himalayan marmots in this study exhibited a unique escape strategy compared with golden marmots and Alpine marmots in that they treated humans as a greater threat than dogs. The running or walking of the threat did not affect the escape reaction of marmot populations experiencing little or no disturbance. These observed diversifications reflect a complex interaction between habituation, perceived threat levels, and visual cues. To further explore the effect of dogs on the escape reaction of Himalayan marmots, different-sized dogs with varied colors should be considered in future studies.

## Figures and Tables

**Figure 1 animals-15-00935-f001:**
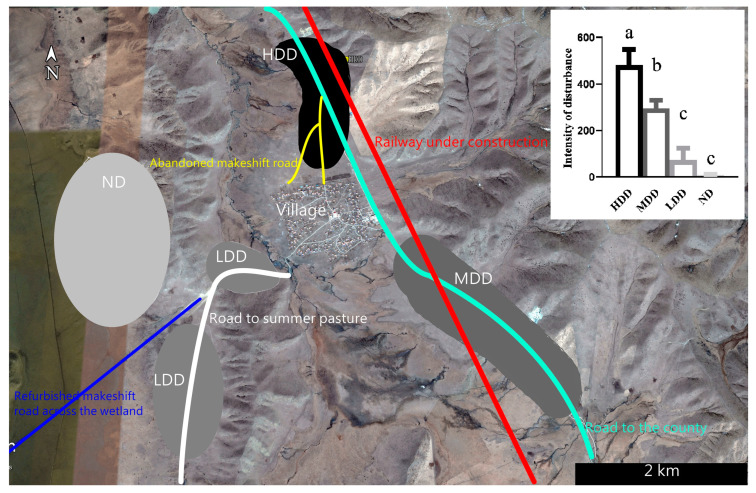
The location of the village and four newly identified regions with different disturbance intensities. Compared with a previous study [7], the high disturbed district (HDD) in this study had a certain area overlap with the previous study’s high disturbed district; the medium disturbed district (MDD) was previously the low disturbed district; and the low disturbed district (LDD) and the natural district (ND) had an area overlap with the previous natural area. The histogram in the upper right corner is the disturbance intensity of vehicle in different districts (different lowercase letters indicate significant differences between groups).

**Figure 2 animals-15-00935-f002:**
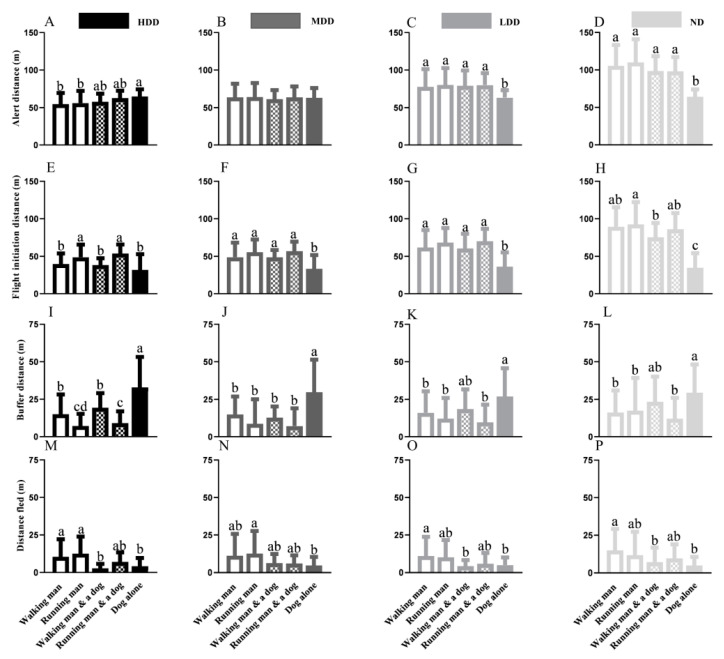
Intra-population variations in alert distance of marmots from HDD (**A**), MDD (**B**), LDD (**C**), and ND (**D**) under different stimuli (a walking man, a running man, a walking man with a leashed dog, a running man with a leashed dog and a single dog); Intra-population variations in flight initiation distance of marmots from HDD (**E**), MDD (**F**), LDD (**G**), and ND (**H**) under different stimuli; Intra-population variations in buffer distance marmots from HDD (**I**), MDD (**J**), LDD (**K**), and ND (**L**) under varied stimuli; Intra-population variations in distance fled of marmots from HDD (**M**), MDD (**N**), LDD (**O**), and ND (**P**) under different stimuli (different lowercase letters indicate significant differences between groups).

**Figure 3 animals-15-00935-f003:**
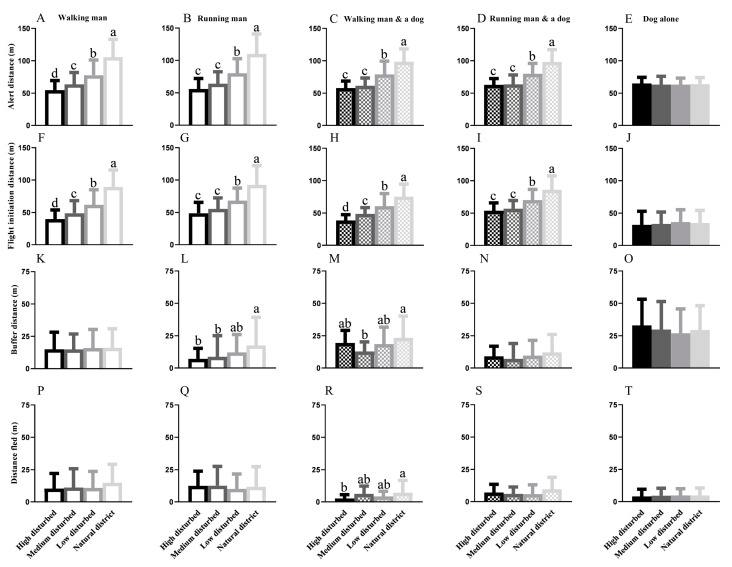
Inter-population variations in alert distance of marmots from four habitats when disturbed by the same stimulus: a walking man (**A**), a running man (**B**), a walking man with a leashed dog (**C**), a running man with a leashed dog (**D**) and a single dog (**E**); Inter-population variations in flight initiation distance of marmots from four habitats when disturbed by a walking man (**F**), a running man (**G**), a walking man with a leashed dog (**H**), a running man with a leashed dog (**I**) and a single dog (**J**); Inter-population variations in buffer distance of the marmots from four habitats when disturbed by a walking man (**K**), a running man (**L**), a walking man with a leashed dog (**M**), a running man with a leashed dog (**N**) and a single dog (**O**); Inter-population variations in distance fled of the marmots from four habitats when disturbed by a walking man (**P**), a running man (**Q**), a walking man with a leashed dog (**R**), a running man with a leashed dog (**S**) and a single dog (**T**) (different lowercase letters indicate significant differences between groups).

## Data Availability

The datasets generated during and/or analyzed during the current study are available from the corresponding author or first author upon reasonable request.

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
