# Peer review of "Effects of Varied Stimuli on Escape Behavior Diversification of Himalayan Marmots for Different Human Disturbances"

_animals, 2025, doi:10.3390/ani15070935_

Round 1
Reviewer 1 Report (Previous Reviewer 3)
Comments and Suggestions for Authors
I reviewed an earlier draft of this paper and faulted it for lack of details and for lack of explanation of its significance--it is a small study and not obviously essential to the world. This revised ms is much improved from the previous version, by including essential data about the actual procedures and context, and by adding details and rigor to the account and discussion. It is still not an earth-shaking finding, but we now know it was conducted with some comprehensiveness and rigor, and is an interesting data paper. As a reviewer I am glad to see my suggestions taken seriously and addressed.
Author Response
I reviewed an earlier draft of this paper and faulted it for lack of details and for lack of explanation of its significance--it is a small study and not obviously essential to the world. This revised ms is much improved from the previous version, by including essential data about the actual procedures and context, and by adding details and rigor to the account and discussion. It is still not an earth-shaking finding, but we now know it was conducted with some comprehensiveness and rigor, and is an interesting data paper. As a reviewer I am glad to see my suggestions taken seriously and addressed.
Answer: Thank you for your valuable suggestion on the earlier edition of the manuscript, otherwise, we won't be able to improve this study. Our subsequent work on seasonal variations in their escape behavior may be more attractive.
Reviewer 2 Report (Previous Reviewer 2)
Comments and Suggestions for Authors
This manuscript is a revision of a previously submitted manuscript. It describes a manipulated experiment of the responses of Himalayan marmot’s to various anthropogenic-related cues. They have attempted to redo a previously published study at the site on the same study species. They found that habituation plays a role in the distances at which marmots flee and recommend that future studies evaluate clothing color and dogs for similarly conducted experiments.
The terminology used is very confusing. FID has very specific terminology (Starting Distance, Alert Distance, and Flight Initiation Distance). The creation of new ones mixed with the previous ones is inappropriate and confusing. I recommend the authors stick to the conformed FID terminology. It will help the manuscript be read by a wider audience and subsequently cited in the FID literature. For example line 368 – “Distance fled….” Gives a wrong connotation to what the authors really mean to say.
Strongly recommend, as in the previous review, to change Flee to Flight throughout the manuscript.
Line 64 – FID must be spelt out before you mention the acronym
The English still needs to be edited – examples line 51 - “A whole day Daily records from 8:00 AM waswere …”
Lines 276-279 – Rewrite the sentence.
Line 280 – coverture
Line 345 – dressings? Reword.
Line 375 – change form to from.
Comments on the Quality of English Language
The authors must make an effort to edit the manuscript. There are too many glitches that hamper the flow of the paper and should be corrected.
Author Response
This manuscript is a revision of a previously submitted manuscript. It describes a manipulated experiment of the responses of Himalayan marmot’s to various anthropogenic-related cues. They have attempted to redo a previously published study at the site on the same study species. They found that habituation plays a role in the distances at which marmots flee and recommend that future studies evaluate clothing color and dogs for similarly conducted experiments.
The terminology used is very confusing. FID has very specific terminology (Starting Distance, Alert Distance, and Flight Initiation Distance). The creation of new ones mixed with the previous ones is inappropriate and confusing. I recommend the authors stick to the conformed FID terminology. It will help the manuscript be read by a wider audience and subsequently cited in the FID literature. For example line 368 – “Distance fled….” Gives a wrong connotation to what the authors really mean to say.
Answers: Thank you for your constructive comments on my manuscript. AD and FID are two most important parameters in such researches, but another two distances (DF and BD) we focus on in the present study can greatly supplement and clarify the escape strategy of animals when threatened.
Distance fled is the distance between flight start point and the final safe location where the threatened individual arrived after a flush, especially for animals that escape into caves or burrows.
BD can measure the difference between the time they start to alert and the time they start to escape.
AD can measure their alertness, while BD can well measure their sensitivity.
We believe that the introduction of DF and BD in relevant research can better explain the problem.
As for the definition of DF in this paper, we have made a more detailed explanation.
Strongly recommend, as in the previous review, to change Flee to Flight throughout the manuscript.
Answers: We appreciate this valuable suggestion. We accepted the proposal and replaced "Flee" with "Flight" or "Escape".
Line 64 – FID must be spelt out before you mention the acronym
Answers: Thank you for your suggestion. That's how we arranged it
The English still needs to be edited – examples line 51 - “A whole day Daily records from 8:00 AM was were …”
Answers: Thank you for this comment. We revised it according to your suggestion.
Lines 276-279 – Rewrite the sentence.
Answers: We are very grateful for this precious suggestion. We have rewritten this sentence.
Line 280 – coverture
Answers: Thank you for your suggestion. We have changed it to "cover".
Line 345 – dressings? Reword.
Answers: Thank you for your suggestion. We replaced it with "clothes".
Line 375 – change form to from.
Answers: Thank you for your suggestion. We have revised it.
Reviewer 3 Report (Previous Reviewer 1)
Comments and Suggestions for Authors
The re-submitted manuscript is suitably edited. I appreciate the changes the authors have made, the removal of errors in citations and the editing of the text. Although the text could be improved, it is acceptable for publication in this version. The only comment I have is that Figure S1 is mentioned in the text, but I could not find it in the materials available for the article. In the supplementary materials there are only Figures from the original article. According to the text it should be a picture of a used dog. In my opinion, the photo could easily be part of the manuscript, readers would certainly appreciate it, especially in relation to the discussion concerning the dog. Also, thank you for your detailed responses to my comments.
Author Response
The re-submitted manuscript is suitably edited. I appreciate the changes the authors have made, the removal of errors in citations and the editing of the text. Although the text could be improved, it is acceptable for publication in this version. The only comment I have is that Figure S1 is mentioned in the text, but I could not find it in the materials available for the article. In the supplementary materials there are only Figures from the original article. According to the text it should be a picture of a used dog. In my opinion, the photo could easily be part of the manuscript, readers would certainly appreciate it, especially in relation to the discussion concerning the dog. Also, thank you for your detailed responses to my comments.
Answers: Sorry and thank you for your reminder and suggestion, we uploaded the wrong picture before. In this new version, we uploaded the correct picture (a group photo of man and dog).
Round 2
Reviewer 2 Report (Previous Reviewer 2)
Comments and Suggestions for Authors
This manuscript is a second revision of a previously submitted and revised manuscript. It describes a manipulated experiment to examine the responses of Himalayan marmots to various anthropogenic-related cues. The authors found that habituation plays a role in the distances at which marmots flee and recommend that future studies evaluate clothing color and dogs for similarly conducted experiments.
The authors have tried to improve the manuscript, but the language is still problematic. The authors must have their manuscript edited by a native English speaker or use an appropriate application to correct grammatical mistakes and misused terms.
Although the authors have stated in the response that they have replaced the term “flee” with either flight or escape, they continue to use it (Title in the column at left, in keywords, lines 33, 42, 67, and so on…..).
Line 29 – Italicize the Latin name.
Line 31 – Delete “Then…”
Lines 59, 62, 67, ….., 279, …. - The authors should replace “coming” with “approaching” throughout the manuscript.
Consider changing “single man” to "individual" or something similar.
Lines 384 – 389 – one long, grammatically wrong sentence. Rewrite into smaller sentences.
Comments on the Quality of English LanguageThe authors must have their manuscript edited by a native English speaker or use an appropriate application to correct grammatical mistakes and misused terms.
Author Response
This manuscript is a second revision of a previously submitted and revised manuscript. It describes a manipulated experiment to examine the responses of Himalayan marmots to various anthropogenic-related cues. The authors found that habituation plays a role in the distances at which marmots flee and recommend that future studies evaluate clothing color and dogs for similarly conducted experiments.
The authors have tried to improve the manuscript, but the language is still problematic. The authors must have their manuscript edited by a native English speaker or use an appropriate application to correct grammatical mistakes and misused terms.
Answers: Thank you for your valuable suggestions. We have applied with MDPI for editing services, but the version shown to the reviewers is not polished (incorrect version), so we downloaded the polished version and revised it according to your comments and suggestions.
Although the authors have stated in the response that they have replaced the term “flee” with either flight or escape, they continue to use it (Title in the column at left, in keywords, lines 33, 42, 67, and so on…..).
Answers: Thank you very much for your suggestion. We have changed all the “flee” in the manuscript to “escape”. Line 49, 65, 67, 86, 94, 165, 190, 192-194, 198, 209, 370, 386
Line 29 – Italicize the Latin name.
Answers: Thank you for this comment. We have italicized the Latin name. Line 29
Line 31 – Delete “Then…”
Answers: Thank you for your comments. We have deleted "then". Line 31
Lines 59, 62, 67, ….., 279, …. - The authors should replace “coming” with “approaching” throughout the manuscript.
Answers: Thank you for your suggestion. We have changed the "coming" in the manuscript to "approving". Line 60, 63, 68, 74, 287, 291, 323, 333, 340, 362, 368, 379, 386
Consider changing “single man” to "individual" or something similar.
Answers: Thank you for this valuable suggestion. We have changed all “single man” in the manuscript to expressions such as "individual" and "man alone" or something similar. Line 36, 42, 224-225, 228, 236, 241, 243,319, 353, 387, 389, 392, 394.
Lines 384 – 389 – one long, grammatically wrong sentence. Rewrite into smaller sentences.
Answers: Thank you very much for your suggestions and comments. The sentences here have been changed after the touch up service of MDPI. The version you reviewed before is incorrect, and there are errors in grammar and words. The manuscript you are reviewing now has been polished. Line 402-407
This manuscript is a resubmission of an earlier submission. The following is a list of the peer review reports and author responses from that submission.
Round 1
Reviewer 1 Report
Comments and Suggestions for Authors
The submitted paper presents data on the escape response of Himalayan marmots depending on the type of threat and the effect of disturbance at different sites. This interesting topic has a seemingly well-developed methodology and thus promises results that would complement similar research in other species. However, the processing and presentation of the results is not very clear and has some limitations. The text gives the impression that it was either completed in a hurry or AI was not very successfully used to produce some parts of the text. It is a shame, because the data collected is quite extensive and the results (when the reader gets to know them) very interesting. I make specific comments on particular parts of the text below.
The introduction as a whole appropriately introduces the reader to the topic, explains the abbreviations and what the paper builds on. On the other hand, you state that various escape strategies and such are often studied and then provide a single citation. The same problem is then present in the discussion. Furthermore, the first sentence of the third paragraph has a citation with the number 8, and the following sentence has the same citation. It is better to give the citation once, preferably at the end of the second sentence, or to choose a different citation here if appropriate. In the next section you give examples of other species in which a similar phenomenon is being studied. It is not clear to me why examples are given for birds in particular when the article focuses on mammals. I understand that sometimes there are no similar studies on mammals, but on the other hand the response to a predator may look different in a small songbird than in a terrestrial mammal. The terrestrial reptile examples make more logical sense from this perspective. I don't understand why the paragraph on line 98 starts by saying that the dog is a domesticated wolf. Other than being trivial information, it's unrelated to the rest of the text. I would have appreciated more information about the local situation regarding domestic dogs. There is no indication of what the coexistence of humans and dogs is like in the study area. By this I mean whether dogs are kept as pets, serve as working animals (e.g. property protection) or whether it is common for them to wander around the village and can pose a risk to wildlife. It would also be useful to add information on commonly bred dog breeds or whether they are more likely to be hybrids of different breeds.
I have a bigger problem with the methodology. While the first chapter (Study site) provides relevant information at first sight, I have no idea, for example, how the citation 25 relates to the location of the village in the study site; GPS coordinates would be more logical. The third sentence refers to a previous study from this site, however, citation 14 (Williams, D.M.; Nguyen, P.-T.; Chan, K.; Krohn, M.; Blumstein, D.T. High human disturbance decreases individual variability in skink escape behavior. Current Zoology 2020, 66, 63-70) was conducted in French Polynesia on lizards. Is this a typo? Again, the same citation is repeated in a relatively short paragraph. Figure 1 is well chosen, only the caption should briefly explain what the abbreviations at each location (HDD, MDD, LDD, ND) mean to make the caption self-explanatory. In the next chapter (missing a capital letter in the title), I again don't understand the first citation. You state that the information was used based on previous studies, then you cite one where I'm not sure if the choice is correct. In the second paragraph the font is different sizes. On the other hand, I do appreciate the pilot study that was used to categorize the sites based on the level of potential disturbance to marmots. For Figure 2, consider modifying the X-axis labels, this way it is not clear, again, it would be useful to explain the abbreviations, and it is not clear what the letters a-c for the F graph mean. From the next chapter (Data collection) I am not able to deduce how much data was collected. In my opinion, the last sentence should be at the beginning, then it should be better explained how much data was collected in which location. There is no information on the time of day when the collection was carried out. In the next chapter (Experimental procedure) the information is better elaborated, only on line 178 a parenthesis is missing. It would be useful to add information about the breed of dog, why it was chosen and what training or qualifications the dog had for this experiment.
In the results section I have first the comments on Figures. It would be useful to unify the style of both Figures, for example to keep the whole Figure in black and white. On the contrary, the different shades in Figure 3 are quite confusing, in Figure 4 they make more logical. Again, the use of lower case letters (a-d) or their absence is missing in both explanations. It will take quite a bit of work for the reader to decode the information. In Figure 3, the captions unnecessarily list the types of disturbance, as they are included in the full text of the X-axis; on the other hand, it would be useful to repeat what the abbreviations for locations mean. The text of this chapter is very difficult to read. I understand that the results are calculated based on a larger number of comparisons using simpler tests and more complex multivariate analysis is not used. However, navigating through the large quantity of similar information is quite difficult. Please consider whether the results of some of the comparisons could be summarised in a table and the most significant results commented on in the text. Or clearly summarise the most significant outcome in one sentence at the end of the paragraph.
The biggest problem I have with the discussion. It's basically a summary of your results and a reflection on why it worked out that way. The most common mention is that it was a small white dog that was hard to see in the grass and uncharacteristic of the environment. The question then arises, why was this dog chosen? Was it a typical representative of the domestic dog that people in the area keep? Otherwise, I don't understand this choice. Further, there are few other papers in the text that your results are compared to, it is mostly a statement that it came out differently/same in another paper and that is all. Again, you give the wording "Different from all other studies" (line 337) followed by a single citation, but this is not a review and the research is conducted on ungulates.
Your research focuses on an interesting topic and a large amount of data has been collected, although it is possible that an inadequate type of stimulus was chosen. Nevertheless, I would like to see the results published, so I recommend the paper be revised and not rejected.
Reviewer 2 Report
Comments and Suggestions for Authors
The authors present a study of how different human disturbances affect the flight responses of Himalayan marmots (Marmota himalayana). They emphasize the impact of various stimuli, including humans and dogs, on the animals' behavior and highlight the importance of understanding wildlife adaptations to human activities.
This is a lengthy manuscript; I recommend shortening it to a Short Communication.
The title with the word “flee” is awkward. Rewrite with either escape/avoidance/alarm/flight. Suggest rewording as “Effects of Varied Stimuli on Escape Behavior Diversification in Himalayan Marmots under Human Disturbance.”
This is also true for the whole manuscript, where the word “flee” has impeded the flow of the text.
Occasionally, grammatical errors or awkward phrasing could be refined for better flow. For example, some sentences may lack parallel structure or have misplaced modifiers. Some ideas are repeated in slightly different forms, which could be streamlined to improve conciseness.
While the authors mention data collection across different populations, more information on sample sizes and the specific characteristics of these populations would strengthen the findings.
Line 78 – easier? Reword.
Figure 2 – see no point in the graphs when they are averaged for all four study areas. Might as well give only average for all together.
Line 180 – change “in the telescope” to “with a telescope. Also, you can specify the brand and magnification of the telescope.
Line 180 – gingerly – please use a scientific term.
Line 405 – delete surprisingly.
Comments on the Quality of English Language
Authors must improve the text by having it edited for grammar and correct word usage. The English must also be corrected to improve the flow of the manuscript.
Reviewer 3 Report
Comments and Suggestions for Authors
This ms is interesting, very well done, well written, and well laid out, but a study based on one person and one small dog is simply not worth publishing. We really need a variety of dogs and people, in a variety of clothing, to get any results worth attention. I would certainly encourage the authors to keep on with the research, but get a respectable sample of dogs--say, 20 or 30 dogs of different size and color.